# VERVE: Template-based ReflectiVE Rewriting for MotiVational IntErviewing

**Do June Min[1], Verónica Pérez-Rosas[1], Kenneth Resnicow[2], and Rada Mihalcea[1]**

[1]Department of Electrical Engineering and Computer Science, [2]School of Public Health
University of Michigan, Ann Arbor, MI, USA
{dojmin, vrncapr, kresnic, mihalcea}@umich.edu

## Abstract

Reflective listening is a fundamental skill that counselors must acquire to achieve proficiency in motivational interviewing (MI). It involves responding in a manner that acknowledges and explores the meaning of what the client has expressed in the conversation. In this work, we introduce the task of counseling response rewriting, which transforms non-reflective statements into reflective responses. We introduce VERVE, a template-based rewriting system with paraphrase-augmented training and adaptive template updating. VERVE first creates a template by identifying and filtering out tokens that are not relevant to reflections and constructs a reflective response using the template. Paraphrase-augmented training allows the model to learn less-strict fillings of masked spans, and adaptive template updating helps discover effective templates for rewriting without significantly removing the original content. Using both automatic and human evaluations, we compare our method against text rewriting baselines and show that our framework is effective in turning non-reflective statements into more reflective responses while achieving a good content preservation-reflection style trade-off.

## 1 Introduction

During the Covid-19 pandemic, the number of people living with anxiety and depression rose more than four times, thus aggravating the ongoing disparity between unmet needs for mental health treatment and increased mental health disorders (Coley and Baum, 2021).

One driving cause behind this discrepancy is the shortage of mental health professionals, which is exacerbated by the fact that becoming a counselor requires extensive training (Lyon et al., 2010). In particular, counselor training is difficult to speed up due to several factors, such as the need for expert supervision, and the laborious and time-extensive process needed to provide evaluative feedback. There

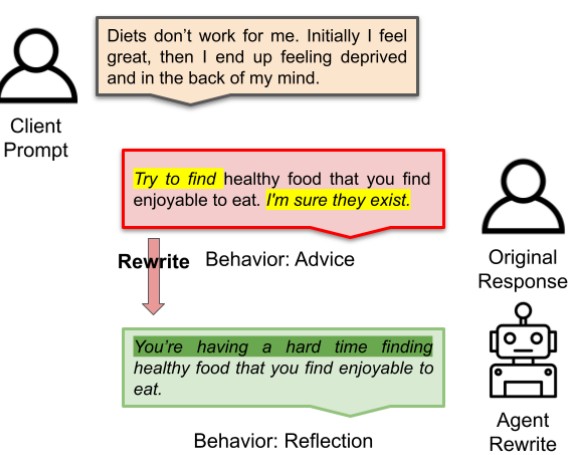

Figure 1: In this example of counselor response rewriting, a counseling trainee is asked to provide a reflective response given the client prompt and produces a poor response by giving a piece of advice rather than reflecting the client's concerns. Our system generates an improved response that preserves content and increases the use of reflective language.

have been several efforts to use NLP to assist counselor training, including automatic coding of counselor behavior (Flemotomos et al., 2021b), providing timing and language suggestions during client interactions (Miner et al., 2022; Creed et al., 2022), and evaluating the quality of specific counseling skills (Shen et al., 2020; Min et al., 2022).

However, the progress in developing tools that can fulfill a "mentoring role" and offer alternative language suggestions for counselors in training has been limited. To fill this gap, we introduce the task of counselor response rewriting, which involves rephrasing trainees' responses with basic counseling skills into alternative responses that reflect a more advanced level of counseling proficiency. We focus on reflective listening as our main counseling skill, and on Motivational Interviewing (Miller and Rollnick, 2013) as the counseling strategy.

We show an example of our system output in Figure 1. In this case, providing a numerical score

or a reference reflection (i.e., a high-quality reflection) does not help the counselor understand what parts of their answer could be improved. Our system addresses this shortcoming by separating the behavior-relevant (e.g., reflection-like language) and the behavior-non-relevant parts, and using the latter as a template for creating an improved rewrite of the original.

We introduce VERVE (ReflectiVE Rewriting for MotiVational IntErviewing), a framework based on template editing methods from text style transfer that do not require parallel data, since expert annotation of rewritten responses is expensive and time-consuming. We propose two simple techniques to adapt template-based text rewriting to the counseling domain: paraphrase-augmented training, and adaptively template updating. The first helps the text generator to learn a more flexible mapping between a masked template and a full response so that the structure of the final rewrite is not constrained by the template. The second handles the content-edit trade-off (e.g., preserving part of the user response rather than completely rewriting) by iteratively updating the masked template based on the effect of the rewrite. We evaluate our framework against several baselines from previous text style transfer works using automatic evaluation and demonstrate that our system outperforms baselines in achieved reflection scores while still preserving content from the original response.

## 2 Related Work

Our work builds upon previous work in text style transfer, text rewriting, and NLP for counseling.

Broadly, counselor response rewriting is related to text rewriting in NLP, which includes, text style transfer, content debiasing, and controlled generation (Li et al., 2018; Madaan et al., 2020). In this work, we focus on rewriting through template-based editing (or prototype-based in other text style transfer literature (Jin et al., 2022). These systems offer several advantages over alternative frameworks such as latent style transfer or LLM-based methods (Dai et al., 2019; Sharma et al., 2023). First, template-based editing systems offer high interpretability as they rely on predefined templates or patterns. Users can have precise control over the editing process by selecting specific templates or designing new ones. This allows for easier understanding and manipulation of the output, which is particularly important in applications where trans-

parency is valued. Moreover, content preservation is another advantage of prototype-based editing, since the template generation process can be controlled to vary the amount of original content preserved in the rewrite. An important difference from previous studies is that we address text rewriting in dialog context, whereas previous studies are mostly concerned with transforming isolated text, such as product reviews (Mir et al., 2019).

Since counseling reflections often include empathy (Lord et al., 2014), empathetic text generation and rewriting are also relevant. While most of the empathetic generation literature focuses on modeling emotion for generating responses from scratch, Sharma et al. (2021) directly models multiple aspects of empathy and applies reinforcement learning (RL)-based training for rewriting online mental health comments. Similarly, we leverage a classifier model for discriminating attribute labels for text but use simple supervised learning instead of policy gradient RL training.

Our work is also related to recent work on NLP for the counseling domain aiming to assist counselors during their practice and ongoing training. Reflection is an important construct in counseling strategies such as MI, and previous works have studied how the frequency or quality of reflections can be used to evaluate counseling (Pérez-Rosas et al., 2017; Flemotomos et al., 2021a; Ardulov et al., 2022). There also have been studies on generating reflections (Shen et al., 2020, 2022). However, to the best of our knowledge, our work is the first to consider rewriting non-reflections into reflections, .

## 3 Counselor Response Rewriting

### 3.1 Task and Application

Reflection is a key skill for empathetic listening in motivational interviewing (Miller and Rollnick, 2013; McMaster and Resnicow, 2015; Moyers et al., 2016a). Recently, there has been increasing interest in how language models can be used to understand and generate reflections to assist counselor practice and ongoing training (Flemotomos et al., 2021a; Shen et al., 2020, 2022). Our work follows the same research direction, however, we focus on the new task of reflection rewriting rather than reflection writing from scratch. We argue that response rewriting can provide more detailed feedback while coaching and training counselors, since users' responses are considered by the model, al-

| Statistics | PAIR | AnnoMI |
|---|---|---|
| # of Exchange Pairs | 2544 | 450 |
| Avg # of Words | 32.39 | 39.50 |
| # of Complex Reflection | 636 | 0 |
| # of Simple Reflection | 318 | 0 |
| # of Non-Reflection | 1590 | 450 |

Table 1: Annotation statics for PAIR and AnnoMI datasets.

lowing the user to compare the original and rewritten responses.

For example, given a client prompt describing their struggles while losing weight, a poorly made counselor response such as "Are you sure *you've given up all unhealthy food*?" contains unsolicited advice rather than listening and acknowledging the client's experience. Given this response, our system can suggest the following rewrite "*You've given up all unhealthy food*s and you're sure that dieting doesn't work for you." as an alternative higher quality reflection that preserves content from the original response.

## 4 Datasets

We use two publicly available MI datasets from PAIR (Min et al., 2022) and AnnoMI (Wu et al., 2022). While PAIR is a collection of single turn exchanges, AnnoMI is a set of counseling conversations consisting of multiple conversational turns. PAIR contains client prompts along with counselor responses with varying reflection quality levels, including simple and complex reflections or non-reflection. Complex reflections go further than simple reflections (i.e., simple repetition or slight rephrasing of client's statement) by inferring unstated feelings and concerns of the client and are often preferred over simple reflections in MI counseling (Miller and Rollnick, 2013). Note that although PAIR contains multiple responses for a given prompt they were not designed as rewrites, and thus they cannot be used directly as parallel data to train a supervised end-to-end rewriter.

**Preprocessing.** We preprocess AnnoMI to focus on single exchanges between counselors and clients. Also, since AnnoMI does not include annotations for reflection type we use a the subset of utterances labeled as no-reflections only. We extract pairs consisting of a single client turn followed by a counselor non-reflection, with constraints on the length of the utterances to filter out short utterances or disfluencies. We include a more detailed descrip-

tion of the datasets and the filtering procedure we used in Appendix A. The final dataset statistics are shown in Table 1.

## 5 Methodology

Our VERVE framework, shown in Figure 2, is based on a template editing-based approach that does not require parallel data. Below, we describe system details.

### 5.1 Template-based Response Rewriting

VERVE follows a two-step process in which attribute-relevant tokens in the counselor response are first identified and masked. The resulting template, along with the original prompt, are then provided as input to the generator to obtain a rewritten response filled with relevant spans.

**Template Extraction (Masking).** The goal of this step is to create a masked version of the original response to be used by the generator as a template for the rewritten version. We start by training a transformer model to discriminate between the three levels of reflections in the PAIR dataset i.e., non-reflection, simple reflection, and complex reflection.[1] Next, we use the attention scores of the discriminator to identify tokens that contribute to the low reflection level in the original response. Our intuition is that the reflection scoring model has learned to attend to key tokens that are relevant to reflection qualityso, their attention scores can be used to signal token importance. We use the model's penultimate self-attention layer to identify tokens to be masked and then normalize the attention scores across tokens, per attention head. We then apply max-pooling over the heads, obtaining a single attention map $A$ over tokens. Using token type ids, we then zero out the attention scores of the client prompt tokens. This final map is then compared to the average attention score $\tilde{A}$ across response tokens. For each response token $A_i$, we mask it if

$$A_i >= \tilde{A} \tag{1}$$

**Rewriting from Template (Filling).** The next step is to input the resulting template into the generator model. This is a transformer-based encoder-decoder model that receives the concatenation of prompt and template as input, separated by a special token. We train the generator on the original

---

[1]The reflection discriminator achieves ∼83% label accuracy.

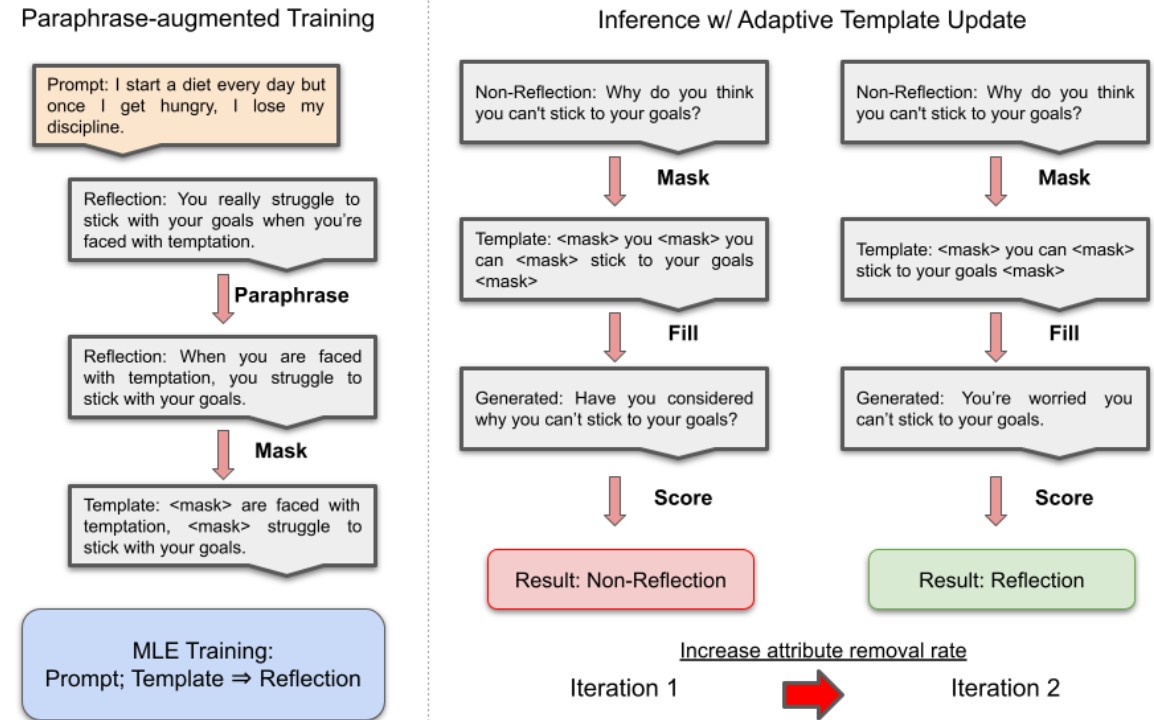

Figure 2: Overview of the VERVE framework. During training, we use attribute-masked versions of paraphrases of reflections as templates for the MLE training for generator training. In the inference time, we adjust the content weight iteratively to achieve the desired edit effect.

response tokens using maximum likelihood estimation (MLE) loss. Importantly, we only use reflections for training the generator (fill-in model).

## 5.2 Paraphrase-augmented Training

One shortcoming we observed while using the template-based editing approach is that it constrains the generator to output responses that are too dependent on the template, thus biasing the generation towards the same type of utterances. For example, "What do you know about yourself?", results in the template: "<mask> do you <mask> about yourself <mask>", with the bigram "do you" biasing the generator towards generating a question rather than a reflection.

To mitigate this problem, we experiment with paraphrase-augmented training, which helps the generator to learn a more flexible mapping between the template and the output by paraphrasing the input template. We use a publicly available transformer-based model (details in A.3) to generate multiple paraphrases for a given response, then we select the version with the highest Levenshtein edit distance from the original response.

## 5.3 Inference with Adaptive Template Updating

One key challenge in text rewriting is the trade-off between content preservation and edit effect since heavy editing of the input (to increase reflection quality/score) leads to less content preservation from the original text. To address this issue, we add a thresholding step during the masking process. Our strategy is similar to Li et al. (2018)'s, who use a tunable thresholding value at test time. We control content masking by weighting the thresholding term $\tilde{A}$ with a weight $C$, using $A_i >= C * \tilde{A}$. Intuitively, higher $C$ values make the content masking more conservative, so only tokens that are highly attended by the predictor would be masked. In contrast, lower $C$ values lead to higher content masking, allowing more room for the generator to fill in.

During inference time, we use $C$ to adaptively adjust the degree of content preservation in the rewrite. We begin with a base $C$ value (e.g., 1.0), and incrementally decrease it (e.g., by 0.1) if the resulting rewrite is not a reflection (or obtains a very low reflection score).[2]

---

[2] To evaluate the quality of the reflection, we use the PAIR scorer (Min et al., 2022).

## 6 Experiments

During our experiments, we use a 75%/5%/20% split of the PAIR data for train, development, and test sets, and use AnnoMI for evaluation only. We report the average scores for five runs based on different random seeds.

### 6.1 Baselines

We compare VERVE against two template-based text style transfer baselines: Delete, Retrieve, and Generate (DRG) (Li et al., 2018) and Tag and Generate (TG) (Madaan et al., 2020). For a fair comparison with our models, we reimplement these baselines using the same base architecture (transformer-based LM) and pretrained weights. We adjust the format of each generator input so they work with a target text and a context prompt in the same way as VERVE. Also, we only implement the template generation methods, to separate implementation details [3] from our comparison and focus on the template generation and filling strategies.

### 6.2 Automatic Evaluations

During our automatic evaluations, we focus on rewriting effectiveness (i.e., whether it leads to a change in reflection score), content preservation, fluency, and relevance.

**Edit Effect (Reflection Score).** We implement the reflection scorer introduced by Min et al. (2022) to measure the reflection quality of the rewrite. The reflection scorer uses a client prompt and a counselor response as input and outputs a scalar value in the range [0,1] measuring the reflection quality in the response. We use the same training and testing split as in our rewriting model so that the test set is unobserved for the scorer. We use the scorer to compute the amount of *change in reflection*.

**Content Preservation.** We are also interested in measuring how much content is preserved in the rewrite, as lower content preservation would reduce the "rewrite"ness of the generation, thus limiting its utility to the user. We use two automatic metrics to measure content preservation: *translation edit rate* and *keyphrase coverage*. Translation edit rate measures the number of changes between the original and rewritten responses. Keyphrase coverage measures how much key information or concepts are included in the rewrite (Snover et al., 2006). We

---

define it as the fraction of keyphrases from the original response found in the rewritten response. We extract keyphrases using the TopicRank algorithm (Bougouin et al., 2013).

**Perplexity, Coherence, Specificity.** Following PARTNER (Sharma et al., 2021), we also measure the perplexity, coherence, and specificity of the rewrite using pre-trained language models.

### 6.2.1 Results

Our evaluation results are shown in Tables 2 and 3. First, we note that VERVE achieves the largest edit effect gain among baselines. We find that while TG performs poorly, DRG shows a higher edit effect, although trailing significantly behind VERVE. VERVE and DRG have similar performances across different metrics, except for changes in edit effect and perplexity. Notably, the two models preserve similar amounts of content, both in terms of keyphrase preservation and edit rate. One interpretation is that VERVE benefits from the editing "room" or space for increasing the reflection score, while DRG uses it to fill in the most likely tokens.

Moreover, we find that although the performances across datasets are slightly different, the general trends are similar, thus indicating that our framework performs well even when applied to unseen data. Finally, we attribute the poor performance of the TG model to the limited size of our corpora. In TG, style markers are selected via salience among $n-$grams found in both corpora. We observe that this choice limits the candidates' size, leading to fewer tokens being masked in the resulting template.

**Ablation Results.** We also perform ablations for the paraphrase augmented training (`paraphrase`), and adaptive template updating (`adaptive`) methods. Across datasets, these methods lead to a higher change in reflection, compared to the base model. Similarly, coherence and specificity are also increased when combining these two strategies. As expected, performance gains for both methods result in lower content preservation.

Interestingly, both methods seem to increase coherence, while paraphrase training is associated with higher specificity. One explanation is that paraphrase training preserves the original text keywords and key phrases. This leads to higher keyphrase coverage (`paraphrase only`) but a lower edit rate result than `adaptive only`.

---

[3]For instance, retrieval from a corpus (Li et al., 2018), or training a separate tagger.

| Model / Metrics | Change in Reflection (%) ↑ | Keyphrase Coverage (%) ↑ | Edit Rate (%) ↓ | Perplexity ↓ | Coherence (%) ↑ | Specificity (%) ↑ |
|---|---|---|---|---|---|---|
| VERVE | **79.86** | 44.30 | 101.33 | 36.97 | **93.35** | 79.21 |
| adaptive only | 44.58 | 51.84 | 50.23 | 36.66 | 82.18 | 73.15 |
| paraphrase only | 49.63 | 63.69 | 76.47 | 39.43 | 81.89 | 79.04 |
| base VERVE | 17.02 | **73.68** | **29.25** | 37.33 | 75.69 | 73.82 |
| DRG ([Li et al., 2018](#)) | 44.56 | 43.37 | 114.56 | **20.82** | 91.06 | **79.81** |
| TG ([Madaan et al., 2020](#)) | 14.66 | 16.80 | 86.11 | 72.09 | 85.62 | 74.43 |

Table 2: Evaluation results for the PAIR dataset. ↑ indicates higher score is better, ↓ otherwise.

| Model / Metrics | Change in Reflection (%) ↑ | Keyphrase Coverage (%) ↑ | Edit Rate (%) ↓ | Perplexity ↓ | Coherence (%) ↑ | Specificity (%) ↑ |
|---|---|---|---|---|---|---|
| VERVE | **74.71** | 34.44 | 87.69 | 34.66 | **91.46** | 74.85 |
| adaptive only | 43.03 | 37.58 | 55.63 | 35.40 | 83.04 | 70.71 |
| paraphrase only | 40.87 | 48.03 | 69.14 | 33.90 | 81.28 | 72.14 |
| base VERVE | 12.81 | **58.31** | **43.90** | 32.39 | 73.26 | 70.93 |
| DRG ([Li et al., 2018](#)) | 48.60 | 26.58 | 96.49 | **18.43** | 88.24 | **75.15** |
| TR ([Madaan et al., 2020](#)) | 6.93 | 17.14 | 90.84 | 60.60 | 76.44 | 71.58 |

Table 3: Evaluation results for the AnnoMI dataset. ↑ indicates higher score is better, ↓ otherwise.

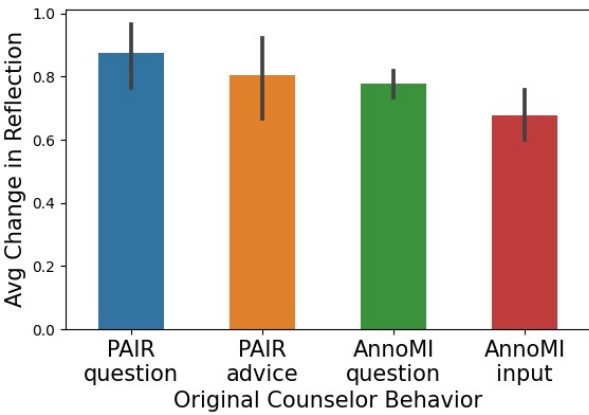

Figure 3: Analysis of edit effect by original counselor behavior. The error bars are 95% confidence intervals.

We also observe interesting differences when comparing paraphrase only and DRG, and adaptive only and DRG. In these comparisons the edit effect results are similar but the content preservation scores are far apart. Overall, adaptive only and paraphrase only are better at preserving content from the original response while achieving a similar edit effect as DRG. This suggests that our framework provides an effective way to explore the trade-off between content and edit effect.

**Analysis by Original Reflection Level.** Additionally, we analyze the edit effect and keyphrase coverage results by the reflection level of the original counselor response, using PAIR's annotation of reflection levels (Complex, Simple, and Non-reflection). In Table 4, we see that VERVE improves reflection scores for no-reflection (NR) and simple reflections (SR). Although edit effect gains decrease for SR (since it has less room for improvement due to already being a reflection), the absolute reflection level is similar for both levels (0.88, 0.87), indicating that VERVE can handle user inputs of varying qualities. In addition, we observe that poorly performing models (TG) can actually reduce the reflection quality of responses. Moreover, we observe that keyphrase coverage is greater for simple reflections. Intuitively, this is likely due to the fact that simple reflections already contain words and spans that also appear in the original response.

**Analysis by Original Counselor Behavior.** We also analyze the changes in reflection quality of the rewrite given the counselor behavior in the original response. We use PAIR's counselor annotations for no-reflections, including "advice" and "question" and AnnoMI annotations for "question", "therapist input", and "other's". From Figure 3, we see that VERVE performs better for "therapist input" than for "question" across both datasets. This suggests that in response rewriting, it is beneficial to consider the dialog act of the original utterance.

|  | Change in Reflection (%) | | | Keyphrase Coverage (%) | |
| --- | --- | --- | --- | --- | --- |
|  | NR | SR |  | NR | SR |
| VERVE | 78.69 | 32.08 | VERVE | 35.95 | 60.48 |
| DRG | 48.13 | 23.56 | DRG | 30.91 | 62.85 |
| TG | 10.28 | -15.52 | TG | 15.48 | 21.51 |

Table 4: Analysis by Original Response Level, on PAIR + AnnoMI dataset. NR and SR refer to non-reflection and simple reflection.

## 6.3 Human Evaluation

For our human evaluation, we consider both experts (counseling coaches, counseling trainees) and non-experts (clients) users. Although our framework is intended for training counselors, we also evaluate with non-experts to ensure that our system can create reflections that sound fluent and empathetic to clients, who are non-MI experts.

**Non-expert Evaluation.** To conduct a human evaluation of the models by non-expert users, we recruit four graduate students without expertise in mental health counseling or motivational interviewing. This setting is intended to evaluate our system from the perspective of counseling patients, who are not experts in MI. To this end, we sample a collection of 50 rewrites, each generated by VERVE and the two baselines, and ask the participants to compare the generations of VERVE against baselines, across four dimensions: fluency, coherence, specificity, and empathy level. We allow no ties during the annotation process.

**Expert Evaluation.** We also evaluate our system from the perspective of MI experts. To this end, we recruit two MI experts (professional MI coaches) to evaluate model rewrites against MI-expert reflections. We use a set of 46 parallel samples from PAIR test split set. Annotators were asked to indicate whether they prefer "A, B, or Tie" when randomly shown reflections either written by our models (rewrites) or by MI experts in response to a given prompt. For a fair comparison of the model and human experts, original responses were hidden. During our evaluations, we opt for comparing VERVE and DRG only against MI experts to mitigate annotation expenses.

We also conducted comparisons between rewrites and expert-written complex reflections using text similarity measures, such as BLEURT, Meteor, BLEU, and BERTScore (Sellam et al., 2020; Banerjee and Lavie, 2005; Papineni et al., 2002; Zhang et al., 2020). For this analysis, we use the full test split of the PAIR dataset. Since the reflections used in this evaluation are not previously seen by our models, the degree of similarity is an indication of how closely model rewrites resemble MI expert reflections.

### 6.3.1 Results

**Non-expert Evaluation** Results for the A/B testing comparison of the models are shown in Table 5. We measure the fraction of times VERVE is preferred against each baseline. Similar to automatic evaluation results, VERVE significantly outperforms TG while having a smaller gap over DRG. It is notable that VERVE surpass DRG on fluency and specificity, while in automatic evaluation DRG outperforms it in perplexity and specificity.

Our evaluation focuses on comparing the quality of the rewritten samples, rather than evaluating whether the generations are indeed rewrites for the original response. Annotation of the usefulness or faithfulness of rewrites is difficult and subject to individual preferences or variations. Overall, we observe that VERVE maintains competitive or higher rates of content preservation while outperforming the baselines in edit effect and conversational quality on both automatic and non-expert evaluation thus showing its potential for response rewriting.

**Expert Evaluation** Results for our A/B testing of the rewriting models against MI experts are shown in Table 6. Unsurprisingly, reflections rewritten by MI experts are generally preferred over model generations. Nonetheless, we find that VERVE is more frequently preferred over experts when compared to DRG by a large margin (36.96% vs 18.48%). Additionally, in a one-to-one comparison our model outperforms DRG with a win rate of 68.48% and a tie rate of 8.70%. Moreover we found that VERVE is most similar to MI expert reflections by a large margin in all metrics, as shown Table 7. These results indicate that our system is capable of producing more expert-like reflections than the baseline models.

| Comparison | | Fluency | Coherence | Specificity | Empathy |
|---|---|---|---|---|---|
| Against DRG (%) | | 61.5 | 56.5 | 58.5 | 62.0 |
| Against TG (%) | | 87.5 | 84.0 | 87.0 | 90.5 |

Table 5: Human comparisons of VERVE vs Baselines using fluency, coherence, specificity, and empathy. The percentages indicate the ratio of VERVE win against respective baselines.

| Model | | Win (%) | Lose (%) | Tie (%) |
|---|---|---|---|---|
| VERVE | | 36.96 | 54.35 | 8.70 |
| DRG | | 18.48 | 73.91 | 7.61 |

Table 6: Model vs MI expert reflections. The percentages indicate the ratio of model wins against MI experts.

| Metric | | Meteor | BLEU | BERTScore | BLEURT |
|---|---|---|---|---|---|
| VERVE | | 67.30 | 35.47 | 58.25 | -29.04 |
| DRG | | 50.23 | 19.84 | 44.58 | -54.04 |
| TG | | 27.29 | 3.61 | 37.00 | -76.73 |

Table 7: Text similarity scores of the different model rewrites against MI-experts.

## 7 Discussion

**Does template editing work for counselor response rewriting?** We argue that template editing is a useful strategy for rewriting counselor responses. However, several adaptations are needed to apply it to the counseling domain. For instance, the prompt should be considered as an additional input and the mask template should be modified accordingly. We found that attention-based token masking is a better fit for response rewriting than $n$-gram-based masking in Madaan et al. (2020); Li et al. (2018) since the relationship between prompts and responses can be naturally modeled by the former. Moreover, it is helpful to model a flexible mapping between templates and reconstructions, because response rewriting may require a greater amount of text editing than style transfer.

**When should rewrites be suggested?** Measuring content preservation and the usefulness of text rewriting are still open problems. However, in the context of counselor response rewriting in MI, we can provide a few guidelines based on our empirical findings. First, we should consider the quality of the original counselor response. Table 4 shows that rewrites of simple reflections have higher content preservation. Second, when rewriting non-reflections, the original intent of the response (counselor verbal behavior) likely matters. In Figure 3, we see that questions have larger edit effects compared to advice or input. Thus, these cases represent situations with greater opportunities for useful feedback. We conjecture that this is due to the overall differences in style and semantics of utterances with different conversational functions and behaviors. For instance, we expect that the directive language in "advice" or "input" responses is more difficult to turn into reflective language.

## 8 Conclusion

In recent years, the disparity between accessible and timely mental health care and the increasing demand for psychotherapy has significantly deteriorated, highlighting the need for more scalable and efficient ways to train new counselors. NLP can assist this counselor training process through automated feedback, which previously was only available through expert supervision.

In this paper, we introduced the task of counselor response rewriting to generate automatic counseling feedback. We introduced VERVE, a template-based approach with paraphrase-augmented training and adaptive template updating, which can transform non-reflective counselor responses into reflective responses. Without access to parallel data, VERVE achieves a higher editing effect than other baseline systems by using flexible template reconstruction approaches. It also has the ability to adjust the attribute masking step without unnecessarily sacrificing content preservation. In future work, we plan to pilot our system in educational settings and explore how VERVE can provide support for student training or coaching.

The VERVE system is publicly available from `https://github.com/mindojune/verve`.

## Limitations

The central intuition behind rewriting as a training or coaching feedback tool is that rewriting can preserve core ideas already present in responses and repurpose them to increase response quality. However, some responses, especially responses containing prescriptive language, may not have salvageable phrases. Although we analyze the impact

of original response behavior on rewriting results, future work on how to identify ideal rewrite opportunities is needed.

Also, measuring content preservation is still an open problem. In this work, we use the fraction of keyphrases in the rewrite as a proxy for measuring content preservation. However, this measure does not fully capture situations where ideas or concepts are expressed in different ways.

Moreover, we note the reflection scorer (PAIR) used in this project is not flawless. We found that the scorer is better at identifying "reflection-sounding" language and often gave high scorers to incoherent or factually wrong responses. Therefore, we used PAIR in conjunction with an evaluation conducted by non-experts and MI experts.

Finally, in this project, we do not consider how large language models (LLMs) can be incorporated as a component in our rewriting framework. Instead, we focus on using smaller, finetunable models that are relatively easier to train, while also being transparent in terms of having components that can be directly observed and examined, such as the attention weights used for template extraction. For future work, we plan to explore how LLMs can augment or complement systems like ours.

## Ethics Statement

We emphasize that our framework is not a tool to process and improve counselor utterances in real counseling practice, nor is it meant to replace or execute the work of human counselors. Rather, it is a tool that focuses on training and coaching learners, and since we find that the generator can make incorrect edits, the rewrites are to be considered only as suggestions. We recommend that in educational deployments safeguards are placed to filter out harmful or toxic edits.

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

# A  Appendix

## A.1  Comparison with Similar Tasks

Here, we briefly discuss how counselor response rewriting is distinguished from related tasks such as text style transfer and empathetic rewriting.

**Text Style Transfer.**   Text style transfer tasks on various styles or attributes have been well-studied in NLP, including sentiment, formality, or toxicity transfer to name a few (Jin et al., 2022). One notable difference between style transfer and response rewriting is that reflection is a verbal strategy that is closer to a dialog act, than a style or sentiment of an utterance. A dialogue act (DA) is defined as an utterance that serves a function in the context of a conversation, such as questioning, making a statement, or requesting an action (Austin, 1962). Commonly used MI coding schemes such as MITI or MISC use DA-like codes such as questions, giving information, etc (Schippers and Schaap , 2005; Moyers et al., 2016a). On the other hand, style transfer is not expected to alter the dialog act of an utterance.

**Empathetic Rewriting.**   Another highly related task is empathetic rewriting, first proposed by Sharma et al. (2021) as the PARTNER system. We first note that as with text style transfer, empathetic rewriting also should not alter the function or dialog act of an utterance. Moreover, PARTNER targets online text-based comments that are usually longer than counseling utterances. Also, although PARTNER can theoretically make fine-grained token-level edits, its edit scope is at a sentence level and largely operates by inserting sentences. Finally, PARTNER focuses on sentence-level edits (removing and adding sentences), and uses a warm-start strategy where a pseudo-parallel corpus is created by identifying high-empathy sentences from the text to create low-high empathy pairs. This is different from our response rewriting since we focus on transforming a relatively short utterance consisting of a few sentences.

### A.1.1  PAIR Dataset

PAIR is a collection of single-turn client-counselor exchanges, collected by (Min et al., 2022). The authors use both expert and crowdsource annotation, using the former for reflection annotation which requires MI expertise, and the latter for collecting non-reflections containing prescriptive language. Following MI literature (Moyers et al., 2016b), each counselor response is coded to one of Complex Reflection (CR), Simple Reflection (SR), or Non-Reflection (NR). Examples are shown in Table 8. CRs are considered higher-quality responses compared to SRs, which are ranked above NRs. In this project, we consider CRs as the gold standard.

| Dataset | Prompt | Response | Label |
|---------|--------|----------|-------|
| PAIR | My mother died of breast cancer, so I know I'm going to die of it too. | Your mother's death was devastating. You're worried you may die the same way she did. | CR |
| | | You believe you will die from breast cancer, just like your mom. | SR |
| | | Are you giving up? | NR |
| AnnoMI | Well, I'd like to see my children settled and my grandchildren growing up and I should be an example to them. | So can it be in there for your -your family's important to you? | NR |

Table 8: Sampled client-counselor exchanges from PAIR and AnnoMI datasets

That is, we aim to rewrite SRs and NRs into CRs.

### A.1.2 AnnoMI Dataset

AnnoMI is a conversation dataset comprising 133 carefully transcribed expert-annotated demonstrations of MI counseling, collected from educational video sources, such as AlexanderStreert[4] (Wu et al., 2022). Although AnnoMI datasets are annotated with session-level counseling quality labels (high or low), we only use utterance-level behavioral codes.

AnnoMI consists of full session-length conversations and is different from PAIR exchanges in several ways. Since the dataset is transcribed from audiovisual sources, it includes many speech disfluencies, repetitions, or interruptions ("Um", "Uh", "I mean–" etc). Thus, we process them to extract single-turn exchanges (client prompt and counselor response).

### A.2 AnnoMI Processing Step

To extract prompt & non-reflection pairs from the AnnoMI dataset, we take the following steps:

1. We flatten the transcripts into consecutive client utterance and counselor utterance pairs.

2. We filter out pairs that meet any of the following criteria:

   - The counselor behavior is not annotated as a reflection.
   - The client utterance string starts or ends with "-". This is to filter out interruptions or continued utterances.
   - We remove common speech disfluencies that we manually identified.
   - The client utterance is shorter than 16 words.
   - The counselor utterance is shorter than 5 words.

---

[4]https://alexanderstreet.com/

### A.3 VERVE Implementation Details

We list the transformer architectures and pretrained weights used for the models used in the project.

- Template Extraction: `bert-base-uncased` (Devlin et al., 2019). We use BERT to leverage its well-trained pretrained weights, but any transformer model that can be trained as a classifier and whose attention weights can be extracted can be used.

- Template Filling: `facebook/bart-large` (Lewis et al., 2019). Our choice of BART as a template filling model is motivated by the fact that BART is trained with a sequence denoising objective, which involves filling in corrupted (masked) tokens.

- Paraphrase Model: `tuner007/pegasus_ paraphrase` https://huggingface.co/ tuner007/pegasus_paraphrase We note that any reasonably well-performing paraphrase model can be used for the paraphrase-augmented training step.

For the sampling algorithm used in the generation of responses, we used beam sampling with num_beams=5.

**Adaptive Template Updating.** At a maximum of 5 iterations, we decrease the content weight $C$ by $0.1$ if the difference is $<= 0.2$.

### A.4 Baseline Hyperparemeters

The baselines tested in this project require setting hyperparameter values for the template creation process. Since the domain of the text is different, we manually tune the threshold parameters by monitoring the reflection score.

- $n-$grams considered: 1,2,3-grams.

- DRG: Threshold: $0.3$.

- TG: $\gamma : 0.75$, threshold: $0.5$.

### A.5 Computational Resources

For training, we use NVIDIA GeForce RTX 2080 Ti for 5 epochs, resulting in a training time of 0.5 hours. We use Pytorch and Huggingface libraries to implement and run our models (Paszke et al., 2019; Wolf et al., 2020).

### A.6 Automatic Metrics Implementation

For the implementation of automatic metrics (perplexity, coherence, and specificity), we follow the implementation in `https://github.com/behavioral-data/PARTNER`.

### A.7 Human Evaluation Details

#### A.7.1 Non-expert Evaluation

Instead of asking participants to evaluate the reflection level of responses, we choose empathy as an evaluation criterion, because the participants are non-experts. MI literature emphasizes that at lower empathy levels, reflective listening is absent, while high empathy is related to skillful use of reflective listening (Miller and Rollnick, 2013). Moreover, the empathetic communication mechanism of "exploration" analyzed in Sharma et al. (2020) is similar to reflection in that both strategies acknowledge the concerns of the client while also actively inferring the client's unstated feelings and expectations.

**Recruitment and Informed Consent.** For our human evaluation, we recruit four PhD students from the department who are in the third or later stages of the degree program. The students are proficient in English and do not have expertise in mental health or MI. They are volunteers and gave informed consent to the usage of their annotation in our project.

**Annotator Instructions.**

- Fluency: How fluent does the response sound?

- Coherence: Is the response a coherent and relevant reply to the prompt?

- Specificity: Is the response specific to the given prompt? (A response is not specific if it can serve as a sensible response to many other prompts.)

- Empathy: How empathetic is the response, given the client prompt?

### A.8 Sample Generations

We show sample generations from VERVE and the baselines in Table 9.

| Prompt | Response (Non-Reflection) | Model | Rewritten Response |
|---|---|---|---|
| My mom said that I can't eat anything after dinner (which is around 5pm) because all that food goes right to your thighs. (PAIR) | Has that been your experience when you've gained weight in the past? | VERVE | You've gained weight in the past and you're concerned that eating after dinner could be one of the reasons why. |
| | | DRG | Since you've lost weight in the past, has that been your experience when you've been trying to lose weight? |
| | | TG | What impact eating has on your thighs has? |
| | | Expert-written | It's frustrating when you get hungry after 5pm but feel like you shouldn't eat anything becuase of your mom's rule. |
| Well, I'd like to see my children settled and my grandchildren growing up and I should, suppose I should be an example to them. (AnnoMI) | So can it be in there for your-your family's important to you? | VERVE | Your family is important to you so you'd like to see it be in there for your family. |
| | | DRG | It's important to you to be a role model for your children and grandchildren, so can it be in there for your-your family's importance to you. |
| | | TG | So much so that your-your family feels important to you? |

Table 9: Sample Rewritings from the models. For the AnnoMI dataset, counselor non-reflections are not paired with parallel reflections.