# OpenReview forum: "VERVE: Template-based ReflectiVE Rewriting for MotiVational IntErviewing"
_EMNLP/2023/Conference — EMNLP 2023 Findings_

### Official Review · Reviewer_AcJk · 2023-08-05

**Soundness:** 4

**Excitement:**

4: Strong: This paper deepens the understanding of some phenomenon or lowers the barriers to an existing research direction.

**Missing References:**

I wished there were more references in the related work section on counseling skills and strategies other than the ones employed in this work and elaborate why reflective listening and motivational interviewing were chosen.

**Paper Topic And Main Contributions:**

The paper describes a template-driven reflective rewriting technique to perform motivational interviewing, which is a counseling strategy that aims to improve counseling outcomes. This is important for scaling efforts in teaching new counselors for mental health support. The proposed technique yields significant improvements to the compared baselines in both human evaluation and automatic evaluation.

**Questions For The Authors:**

- Why is the reflection behavior more preferable than the one that gives advice in Figure 1?
- Why is the penultimate self-attention layer used rather than the final layer? (lines 231-233)
- Is the research on reflective rewriting relatively new? The suggested baselines seem to be not specifically trained for this task and wonder whether there are stronger baselines that VERVE should be compared against.

**Reasons To Accept:**

- The paper is well-written and easy to follow except for the gap on why reflective listening is useful for counseling and what motivational interviewing is.
- Each step of the proposed framework is well-motivated and easy to understand.
- The authors perform a comprehensive analysis for comparing against baselines and expert references using both human evaluation and automatic evaluation.
- The paper has a solid discussions section and a limitations section.

**Reasons To Reject:**

No major reason for rejection other than that the gap on why reflective listening is useful for counseling and what motivational interviewing is. Without this information, readers who are not familiar with the two concepts and their importance to counseling will not fully appreciate the significance of this work.

**Reproducibility:**

4: Could mostly reproduce the results, but there may be some variation because of sample variance or minor variations in their interpretation of the protocol or method.

**Reviewer Confidence:**

2: Willing to defend my evaluation, but it is fairly likely that I missed some details, didn't understand some central points, or can't be sure about the novelty of the work.

**Typos Grammar Style And Presentation Improvements:**

- The introduction should introduce, at least briefly, what motivational interviewing is. It should be explained in more detail in 3.1 when it is heavily referenced by a lot of work (lines 149-151).
- There is a gap in how reflective listening improves counseling outcomes. This should be filled in the introduction.
- (very minor and personal) I'm not a fan of the acronym as it is really forced and also the definition of verve doesn't really resonate with what the model is trying to do. What about Motivational Interviewing with Reflecting Rewriting for COunseling ImpRovement (MIRROR)?
- Table 4 and Table 5 can be column-width tables to reduce space and add more background on counseling skills and strategies and why they are useful for better counseling outcomes.
- line 408: need space after . in ".In Table 4"

---

> ### Author Rebuttal · Authors · 2023-08-28
>
> We appreciate the detailed review and valuable feedback provided by the reviewer. We are glad they found our paper well-written and easy to understand, and our analyses comprehensive. Their insights are very helpful in refining and enhancing our paper.
>
> [Regarding the significance of reflective listening and motivational interviewing] We recognize the importance of providing a clear context for readers unfamiliar with these concepts. We will incorporate a more comprehensive explanation of these concepts in the introduction, detailing why reflective listening is valuable in counseling and elaborating on the principles of motivational interviewing. This will ensure that readers can better appreciate the foundation on which our work is built.
>
> [Regarding Figure 1] We will further elaborate on why reflection behavior is preferable over giving advice by emphasizing the importance of reflective listening in MI. We will also discuss how reflection promotes client autonomy, encourages self-exploration, and aligns with the core principles of MI.
>
> [Regarding the choice of the penultimate self-attention layer] The decision was made by (1) practical advice/wisdom from colleagues and the ML engineering community and (2) our findings that qualitatively the attention scores did seem to correlate with our intuition. We will state this explicitly in the paper.
>
> We acknowledge the point raised about the relatively new nature of research on reflective rewriting, and how different methods and baselines will make our case stronger. In this study, we primarily focused on previous template-based editing methods and aimed to provide improvements over them, but we also recognize the fact that there have been strides in the field of response rewriting, including latent space-based methods and more recently, LLM-based rewriting. Although we have made sure to train the previous template-based baselines on our task using the same model size/training parameters for a fair comparison, we agree that a wider comparison of our methodology will better contextualize the performance of our VERVE model.
>
> In response to this point, we implemented and tested an LLM-prompting-based rewriting approach (in-context learning with instructions and examples of reflections) on the automatic evaluation metrics and also did a qualitative review of the generations. The following are the results:
> PAIR: Change in Reflection: 85.00, Keyphrase Coverage: 13.26, Edit Rate: 246.12, Perplexity: 17.2271, Coherence: 97.58, Specificity: 83.02
> AnnoMI: Change in Reflection: 60.86, Keyphrase Coverage: 18.41, Edit Rate: 131.73, Perplexity: 18.1430, Coherence: 93.64, Specificity: 77.14
> Overall, we observe that the LLM method is capable of producing highly reflective and fluent rewrites, surpassing even the performance of VERVE in the PAIR dataset. However, we also note that the LLM approach results in the most significant departure from the original text.
>
> We posit that LLM rewriting functions within a distinct paradigm, neither superior nor inferior, but different from ours. Our strategy revolves around templates, enabling the model to operate within predefined structures and facilitate incremental adjustments. In contrast, LLM-driven prompts appear to enjoy a higher degree of content addition and removal flexibility. Managing this variance can be challenging, although refining prompts presents a potential solution. We think that a hybrid approach, where templates are provided to the LLM for completion, presents an opportunity we hope to explore subsequently.
>
> Additionally, we think that a locally hosted model for counselor response rewriting can be preferable over LLMs (especially closed-source ones only accessible via API) for privacy concerns and resource efficiency. It ensures that sensitive information is kept in a controlled environment and provides more direct control over computational resources and time constraints.
>
> We appreciate the suggestion to include more references in the related work section related to counseling skills and strategies. We will incorporate additional references that help elaborate on the importance of reflective listening and motivational interviewing, providing a more comprehensive context for our work.
>
> [Regarding the acronym "VERVE,"] we appreciate the reviewer's proposal of "MIRROR." We find it to be a compelling name that better captures the nature of the reflection rewriting task. We are considering this suggestion seriously and will either adopt a name that more accurately resonates with our approach or potentially opt for the reviewer's suggested term.
>
> We will make the necessary adjustments to Table 4 and Table 5 to enhance readability and provide more background on counseling skills and strategies to demonstrate their relevance for improved counseling outcomes.
>
> Finally, we'll address the grammar and style improvements pointed out by the reviewer to ensure the clarity and cohesiveness of the paper.
>
> We thank the reviewer once again for their thorough review and constructive feedback, which will undoubtedly contribute to the overall quality and impact of our paper.

---

### Official Review · Reviewer_DEMb · 2023-08-05

**Soundness:** 2

**Excitement:**

3: Ambivalent: It has merits (e.g., it reports state-of-the-art results, the idea is nice), but there are key weaknesses (e.g., it describes incremental work), and it can significantly benefit from another round of revision. However, I won't object to accepting it if my co-reviewers champion it.

**Paper Topic And Main Contributions:**

Reflective listening is essential for counselors practicing motivational interviewing, as it focuses on understanding and delving deeper into the client's expressions. This study introduces a task named "counseling response rewriting," aiming to transform non-reflective statements into reflective ones. For this, the VERVE framework was created, utilizing template editing techniques from text style transfer. Unique about VERVE is that it doesn't demand parallel data, acknowledging that obtaining expertly annotated rewritten responses is both costly and time-intensive. Two key methods were introduced: paraphrase-augmented training and adaptively template updating. The former encourages the text generator to create a varied connection between templates and full responses, ensuring the end rewrite isn't strictly bound by the template. The latter manages the balance between content editing, focusing on maintaining aspects of the original user response.

The motivation of applying reflective listening is a very important topic in natural language generation. This work proposes an approach that diversity the outcome from a template-based rewrite approach. When compared with previous benchmarks, the proposed framework VERVE showed superior performance in reflection scores while still retaining the original content. However, a notable shortcoming is the absence of comparisons with Large Language Models (LLMs). Given that LLMs have demonstrated proficiency in empathy-aware conversations and paraphrasing, they could serve as potential benchmarks or even offer supplementary techniques in both data generation and the main task of reflective listening. The exclusion of LLMs as a comparative baseline may limit the comprehensiveness of the evaluation, potentially overlooking the advancements and capabilities that these models bring to the domain. Incorporating LLMs in iterations would provide a more holistic view of the landscape and might lead to further optimizations or integrations that could enhance the effectiveness of the proposed framework. In other words, with comparison with LLMs, the contribution of this work can be clearer.

**Reasons To Accept:**

* The study on reflective listening is important for advanced conversational agents.
* This paper is well-written.
* The proposed approach is interesting and show its effectiveness in experiments.


**Reasons To Reject:**

* Given that the study is being conducted in 2023, the motivation and the research background of this work should be more closely aligned to the current state of technology, specifically the advent and impact of large language models. It's essential for research to be contextualized within the most recent and significant developments in the field to ensure its relevance and applicability. As LLMs have shown their ability in generating empathy-aware responses, it is important to check if the kind of reflective listening addressed in this work is still a challenge.
* The paper exhibits a significant oversight by not comparing its techniques with the capabilities of large language models. LLMs have become benchmarks in numerous NLP tasks, including conditional paraphrasing. In this work, LLMs' absence in the comparative analysis leaves a substantial gap in the research. Without this comparison, it's challenging to ascertain the true efficacy and novelty of the proposed methods in light of existing cutting-edge technologies.

**Reproducibility:**

3: Could reproduce the results with some difficulty. The settings of parameters are underspecified or subjectively determined; the training/evaluation data are not widely available.

**Reviewer Confidence:**

4: Quite sure. I tried to check the important points carefully. It's unlikely, though conceivable, that I missed something that should affect my ratings.

---

> ### Author Rebuttal · Authors · 2023-08-24
>
> We appreciate the reviewer's thoughtful evaluation of our paper and their valuable feedback regarding the alignment of our work with the current state of technology and the absence of a comparison with large language models (LLMs).
>
> [Regarding contextualization within the latest technological developments] We acknowledge the importance of considering the impact of LLMs on the field of NLP. While our paper was conceived before the widespread adoption of these models in the field of counseling response generation, we understand that LLMs have indeed made significant strides in generating empathy-aware responses. We will revise our introduction and motivation sections to better situate our work within the context of LLMs and their capabilities.
>
> We also recognize the omission of a comparison with LLMs in our analysis, which could provide a more comprehensive evaluation of our proposed techniques. It is true that LLMs have become benchmarks in various NLP tasks, including response rewriting.
>
> In response to this point, we implemented and tested an LLM-prompting-based rewriting approach (in-context learning with instructions and examples of reflections) on the automatic evaluation metrics and also did a qualitative review of the generations. The following are the results:
> PAIR: Change in Reflection: 85.00, Keyphrase Coverage: 13.26, Edit Rate: 246.12, Perplexity: 17.2271, Coherence: 97.58, Specificity: 83.02
> AnnoMI: Change in Reflection: 60.86, Keyphrase Coverage: 18.41, Edit Rate: 131.73, Perplexity: 18.1430, Coherence: 93.64, Specificity: 77.14
> Overall, we observe that the LLM method is capable of producing highly reflective and fluent rewrites, surpassing even the performance of VERVE in the PAIR dataset. However, we also note that the LLM approach results in the most significant departure from the original text.
>
> We posit that LLM rewriting functions within a distinct paradigm, neither superior nor inferior, but different from ours. Our strategy revolves around templates, enabling the model to operate within predefined structures and facilitate incremental adjustments. In contrast, LLM-driven prompts appear to enjoy a higher degree of content addition and removal flexibility. Managing this variance can be challenging, although refining prompts presents a potential solution. We think that a hybrid approach, where templates are provided to the LLM for completion, presents an opportunity we hope to explore subsequently.
>
> Additionally, we think that a locally hosted model for counselor response rewriting can be preferable over LLMs (especially closed-source ones only accessible via API) for privacy concerns and resource efficiency. It ensures that sensitive information is kept in a controlled environment and provides more direct control over computational resources and time constraints.
>
> We also will include a discussion of recent LLM-based approaches in related tasks (such as cognitive reframing, by Sharma et al https://aclanthology.org/2023.acl-long.555/, and how our approach is related to and different from them.
>
> Thank you for your insightful review. We are committed to addressing these concerns to enhance the relevance, rigor, and impact of our work.

---

### Official Review · Reviewer_LWEi · 2023-08-12

**Soundness:** 2

**Excitement:**

2: Mediocre: This paper makes marginal contributions (vs non-contemporaneous work), so I would rather not see it in the conference.

**Missing References:**

-

**Paper Topic And Main Contributions:**

The paper tackles the problem of response rewriting in the mental health space. The authors exploited two specific techniques involving paraphrase-augmented training and template updating. Further, the results inherit both automatic and human evaluation to show the effectiveness of the proposed system.

**Questions For The Authors:**

The paper raises an important problem to solve. However, the authors have made some assumptions on which I would like to gain clarity. How did the authors segregate between behavior-relevant and non-relevant parts of the dataset? I would like to see some explanation here.

**Reasons To Accept:**

1. The authors targeted an essential problem of counselor training in a very sensitive research space of mental health.

2. The proposed system is a better fit in the current research scope, which targets adapting and rewriting responses rather than generating new ones. This approach is more realistic than a complete replacement of a human therapist.

3. Additional expert validation and ethical considerations are ensured to justify the research standard in this space.

**Reasons To Reject:**

While the paper addresses an important problem related to rewriting responses in the counseling education domain, there are some concerns regarding its novelty and contribution.

1. The literature review shows that ample research already exists in this area. Although the authors present a paraphrase-augmented training basis as their USP, the problem of topic/domain shifting in paraphrase-augmented training is a common concern. The paper fails to discuss the intricacies involved with the approach and "when not to take action" especially in the case of uncontrolled domain shift.

2. Although a mix of expert and non-expert validation has been included, no domain-centric or expert-guided metric is discussed. Being from a very similar field, I believe it is important to involve metrics or methods in psychiatric validation that lead to computational and psychological performance comparison. The question is why authors missed presenting such domain-centric metrics even with the presence of expert evaluators.

3. Despite the existence of ample work, the selection of baselines is very limited to naive/old models. I encourage authors to add more baselines to perform robust performance comparisons.

4. The proposed work lacks novelty. Several methods in the past have tried to apply very similar incorporation to state-of-the-art paradigms and have reported gains in response rewriting space and, in some cases, counseling rewriting too.

**Reproducibility:**

N/A: Doesn't apply, since the paper does not include empirical results.

**Reviewer Confidence:**

5: Positive that my evaluation is correct. I read the paper very carefully and I am very familiar with related work.

**Typos Grammar Style And Presentation Improvements:**

-

---

> ### Author Rebuttal · Authors · 2023-08-24
>
> We appreciate the reviewer's careful consideration of our paper and their thoughtful feedback. We are glad they found our work to be addressing an important problem. We would like to address the concerns raised in the review and provide further insight into our work.
>
> [Regarding the concern about the novelty and contribution of our work]. We acknowledge that there is prior research in the area of response rewriting. Our focus, however, is on the specific context of counseling response rewriting, which has unique nuances due to the requirement for reflective listening in MI. While the concept of paraphrase-augmented training is not entirely novel, we emphasize its application to the counseling domain, which has distinct linguistic and psychological aspects. We agree that domain shifting is a concern, and we have taken steps to address it by utilizing an adaptive template updating mechanism, which we believe mitigates the problem to a degree by incrementally updating the original response.
>
> We agree knowing "when not to take action" is a critical problem. In our previous studies and deployments, our approach to determining the appropriate instances for suggesting rewrites or providing exemplary counselor responses for MI counselor trainers was to employ a reflection detector/scoring model. This model would identify cases where the output fell below a certain threshold, akin to the process that governs the termination of adaptive template updating. However, this approach is a very simple method that may fail in the case of domain shifts, as pointed out by the reviewer. To address this limitation, we have curated a dataset in which each conversation has been annotated with a specific counseling topic, such as eating disorders or smoking cessation. In an upcoming follow-up study, we plan to leverage this dataset to investigate the impact of domain shifts and to devise more effective countermeasures.
>
> [Regarding domain-centric metrics] Our evaluation encompassed a mix of expert and non-expert validation, with our expert assessment aligning with how MI experts gauge and identify reflective listening in practice. However, we agree that further studying the effect of rewritten reflections through psychiatric validation as suggested by the reviewer would further ground our results. We are consulting with our domain expert cooperators about this step, and plan to include additional analyses in the camera-ready.
>
> [Regarding the choice of baselines] We agree that a broader selection of baselines could provide a more comprehensive performance comparison. Our main focus in our work was to compare against template-based editing methods.
>
> In response to this point, we implemented and tested an LLM-prompting-based rewriting approach (in-context learning with instructions and examples of reflections) on the automatic evaluation metrics and also did a qualitative review of the generations. We used OpenAI’s closed-sourced gpt-3.5-turbo-16k-0613 model. The following are the results:
> PAIR: Change in Reflection: 85.00, Keyphrase Coverage: 13.26, Edit Rate: 246.12, Perplexity: 17.2271, Coherence: 97.58, Specificity: 83.02
> AnnoMI: Change in Reflection: 60.86, Keyphrase Coverage: 18.41, Edit Rate: 131.73, Perplexity: 18.1430, Coherence: 93.64, Specificity: 77.14
> Overall, we observe that while LLM method is capable of producing highly reflective and fluent rewrites, at the same time it is also the furthest from the original text thus having the least contextual coherence.
>
> We posit that LLM rewriting functions within a distinct paradigm, neither superior nor inferior, but different from ours. Our strategy revolves around templates, enabling the model to operate within predefined structures and facilitate incremental adjustments. In contrast, LLM-driven prompts appear to enjoy a higher degree of content addition and removal flexibility. Managing this variance can be challenging, although refining prompts presents a potential solution. We think that a hybrid approach, where templates are provided to the LLM for completion, presents an opportunity we hope to explore subsequently.
>
> Additionally, we think that a locally hosted model for counselor response rewriting can be preferable over LLMs (especially closed-source ones only accessible via API) for privacy concerns and resource efficiency. It ensures that sensitive information is kept in a controlled environment and provides more direct control over computational resources and time constraints.
>
> We will include details about our LLM methodology (prompt, model, example selection) in the revision, and also discuss recent methodologies rooted in LLMs within analogous tasks (for instance, cognitive reframing, by Sharma et al: https://aclanthology.org/2023.acl-long.555/). We intend to elaborate on both the connections and distinctions between our approach and these LLM-based strategies.
>
> Concerning the query about segregating behavior-relevant and non-relevant portions of the dataset, we primarily relied on the annotations available within the datasets. In the PAIR dataset, each data sample constitutes a single exchange addressing a counseling topic/issue between a client and a counselor. The AnnoMI dataset comprises counseling dialogues spanning entire sessions. To process this, we identified client-counselor utterance turns concluding with a counselor reflection (identified by the dataset labels). We also filtered out irrelevant behaviors such as disfluencies or interruptions (uh, umm, etc.).
>
> We thank the reviewer for the careful and valuable feedback as it helps us improve the clarity and rigor of our work. We are committed to addressing the concerns raised and enhancing the quality of our paper.

---

### Meta-Review · Area_Chair_JSZg · 2023-09-16

**Recommendation:** 3

**Metareview:**

This paper presents a method for rewriting junior counselor's responses using a system of templates to offer better responses in terms of reflective listening, a key skill for mental health counselors.  The templates themselves are too restrictive, and so paraphrasing is used to broaden their matching power.

One of the reviewers cited several concerns with the lack of comparison with a broader set of relevant methods, and multiple reviewers pointed out the lack of adequate comparison to LLM's.  The paper now argues for why template-based approaches are superior to any LLM based approach on the grounds that they provide more controllability and preservation of the original content.  Controlled generation is an active area of research for LLM's, but an LLM, particularly one that has been fine-tuned, offers quite a high degree of controllability for a given task, and so this claim should be properly evaluated.  Moreover, saying that templates provide a better ability to preserve original content is also a claim that needs evaluation.  The authors note in one of their rebuttals that they have, in fact, tried to use LLM's and that they were superior to their own method, but less good at preserve original text, but this was without sufficient attempts to improve this capability.

This paper explores an important area, is (generally) clearly written and offers good results in its proposed direction.  However, the reviewers here have pointed out legitimate concerns with this work and its place alongside similar work in the area of response rewriting and, indeed, in the broader area of NLP.

---

### Decision · Program_Chairs · 2023-10-07

**Decision:**

Accept-Findings

**Comment:**

This paper presents a method for rewriting junior counselor's responses using a system of templates to offer better responses in terms of reflective listening, a key skill for mental health counselors.  The templates themselves are too restrictive, and so paraphrasing is used to broaden their matching power.

One of the reviewers cited several concerns with the lack of comparison with a broader set of relevant methods, and multiple reviewers pointed out the lack of adequate comparison to LLM's.  The paper now argues for why template-based approaches are superior to any LLM based approach on the grounds that they provide more controllability and preservation of the original content.  Controlled generation is an active area of research for LLM's, but an LLM, particularly one that has been fine-tuned, offers quite a high degree of controllability for a given task, and so this claim should be properly evaluated.  Moreover, saying that templates provide a better ability to preserve original content is also a claim that needs evaluation.  The authors note in one of their rebuttals that they have, in fact, tried to use LLM's and that they were superior to their own method, but less good at preserve original text, but this was without sufficient attempts to improve this capability.

This paper explores an important area, is (generally) clearly written and offers good results in its proposed direction.  However, the reviewers here have pointed out legitimate concerns with this work and its place alongside similar work in the area of response rewriting and, indeed, in the broader area of NLP.